# Distance Correlation-Based Feature Selection in Random Forest

**DOI:** 10.3390/e25091250

**Published:** 2023-08-23

**Authors:** Suthakaran Ratnasingam, Jose Muñoz-Lopez

**Affiliations:** Department of Mathematics, California State University, San Bernardino, CA 92407, USA; 005186749@coyote.csusb.edu

**Keywords:** feature selection, random forest, Pearson correlation, distance correlation

## Abstract

The Pearson correlation coefficient (ρ) is a commonly used measure of correlation, but it has limitations as it only measures the linear relationship between two numerical variables. The distance correlation measures all types of dependencies between random vectors *X* and *Y* in arbitrary dimensions, not just the linear ones. In this paper, we propose a filter method that utilizes distance correlation as a criterion for feature selection in Random Forest regression. We conduct extensive simulation studies to evaluate its performance compared to existing methods under various data settings, in terms of the prediction mean squared error. The results show that our proposed method is competitive with existing methods and outperforms all other methods in high-dimensional (p≥300) nonlinearly related data sets. The applicability of the proposed method is also illustrated by two real data applications.

## 1. Introduction

Feature selection is a crucial aspect of model construction in machine learning. Its main objective is to identify the most significant features while eliminating irrelevant, redundant, and noisy ones. This process involves selecting a subset of the most prominent features. Feature selection is widely used for various reasons, including enhancing model interpretability, reducing learning time, improving learning accuracy, and overcoming the curse of dimensionality, among others. This method is widely employed in many fields, particularly in classification tasks such as bioinformatics data analysis, image recognition, change point detection, and others. Various techniques have been proposed in the literature for evaluating feature subsets in machine learning. The filter method, as described by [1,2], utilizes the intrinsic properties of data to assess feature subsets. The wrapper method, as discussed by [3,4], determines the best subset of features useful for the task based on the performance of the learning algorithm. Finally, the hybrid approach, as described by  [5,6,7], makes use of both filters and wrappers by utilizing independent criteria and learning algorithms to measure feature subsets. Additionally, AIC and BIC criteria are used to identify the ‘best model’. One popular method is the Lasso, which was introduced by [8] and employs ${𝓁}_{1}$ regularized linear regression model. Other Lasso-based feature selection methods have been developed since then, such as Adaptive Lasso [9], Lars [10], and elastic net [11], among others. However, when dealing with high-dimensional data, Lasso methods can face two significant problems: high computational cost and over-fitting. The correlation coefficient (CC) is a criterion, introduced by [12], utilized in feature selection for multiple machine learning algorithms. Ref. [13] used the CC, amongst other measures, for feature selection in high-dimensional data analysis. Ref. [14] made improvements to their models using the CC as well as a clustering technique to filter out less important parameters. We even see [15] use the CC for detecting daily activities in smart homes, where models rely heavily on selecting the appropriate features for these daily activities, and thus on feature selection.

Random forests (RF) is an ensemble learning algorithm that was first proposed by [16]. This method utilizes decision trees and can perform both classification and regression analyses. It achieves this by using a combination of the bootstrap aggregation method and the random subspace method to generate a collection of decision trees, which are then utilized for classification purposes. When building a random forest, the best predictor from a randomly chosen subset of predictors is used to divide each node. Although this method may seem counterintuitive, it has proven to be more effective than other classifiers such as discriminant analysis, support vector machines, and neural networks. Additionally, ref. [16] showed that this approach is resistant to overfitting. According to [17], when a data set has a small number of relevant features and a large number of irrelevant features, RF algorithms may not be able to attain the intended predictive performance, especially if the algorithm selects only a few features at each node. Several methods have been proposed in the literature to improve the performance of the [16]’s traditional RF. For example, ref. [18] establishes consistency of a special type of purely random forest model where strong variables have a larger probability of selection as a splitting variable. Ref. [19] proposed a modification to the standard RF algorithm called Reinforcement Learning Trees (RLT), which involves using a specific type of splitting variable selection and muting of noise variables to prioritize strong variables in the initial stages of tree construction, and gradually decreasing the number of candidate variables towards the terminal nodes. Ref. [20] investigated regression problems within the context of random forest algorithms by focusing on the selection of significant features that are strongly correlated with the response variable. The Pearson product-moment correlation is a criterion to identify features that exhibit high levels of correlation with the response. The Pearson product-moment correlation (ρ) has some drawbacks. One issue is that it only measures the linear relationship between two random variables, *X* and *Y*. Additionally, ρ=0 indicates that *X* and *Y* are independent only if their joint distribution is bivariate normal. Furthermore, even if *X* and *Y* are dependent, the ρ can still be zero.

To remedy this, ref. [21] introduced distance correlation (dCor) that measures all types of dependence between random vectors *X* and *Y* in arbitrary dimensions. The dCor is bounded between 0 and 1, and it equals zero only when the random vectors are independent. According to [22], the dCor is effective in identifying nonlinear relationships that cannot be detected by the Pearson correlation coefficient. Additionally, it can be used for random variables of any dimension, unlike the Pearson correlation coefficient, which is limited to two-dimensional variables. This paper introduces a new approach that incorporates the dCor as a pre-processing step in the conventional RF algorithm for high-dimensional nonlinear datasets. Specifically, we utilize the dCor to select the features that have a significant correlation with the response variable, which are then used in the construction of the RF.

## 2. Main Results

Consider a set of *p* features, $X=(X_{1},\ldots ,X_{p})$, and the dependent variable *Y*. The goal is to estimate the regression function f(x)=E(Y|X=x) and we assume that Y=f(x)+ϵ. We observe a sample of i.i.d. training observations $D_{n}=\left\{\left(X_{1},Y_{1}\right),\left(X_{2},Y_{2}\right),\ldots ,\left(X_{n},Y_{n}\right)\right\}$, where each $X_{i}={\left(X_{i1},\ldots ,X_{ip}\right)}^{⊤}$ denotes a set of *p* variables from a feature space X. Let $\epsilon_{i}$’s be i.i.d. with mean 0 and variance $\sigma^{2}$ and $p^{*}$ refers to the chosen features after removing the ones that have less correlation with the response. The remaining $p-p^{*}$ variables have no influence on the response. We also assume that the expected value $E(Y|X^{*})$ is completely determined by a set of $p^{*}<p$ variables, which means $E\left(Y|X^{*}\right)=E\left(Y|X_{1},X_{2},\ldots ,X_{p^{*}}\right)$.

In their work, ref. [21] proposed a statistical measure called distance correlation (DC) that quantifies all forms of dependence between random vectors *X* and *Y* in arbitrary dimensions, unlike Pearson CC, which is limited to two-dimensional variables. The DC ranges from 0 to 1, and it equals 0 only when the random vectors are independent. According to [22], the DC is effective in detecting nonlinear relationships that cannot be detected by the Pearson CC. The DC is a measure of dependence between two variables that measure the distance between their two characteristic functions. In the bivariate normal case, the DC becomes the Pearson product-moment correlation ρ (CC).

**Definition 1.** 
*Supposing random variables X and Y have finite and positive variances, the distance correlation (R) is defined as,*

$$\begin{array}{c} R\left(X,Y\right)=\frac{\mathrm{dCov}(X,Y)}{\sqrt{\mathrm{dCov}(X,X)\cdot \mathrm{dCov}(Y,Y)}}, \end{array}$$

*where dCov(X,Y) is the distance covariance between random variables X and Y.*


The dCov(X,Y) is defined as follows.
$$\begin{array}{cc} dCov(X,Y) & =\sqrt{\int_{R^{p+q}}\left|\right|f_{X,Y}\left(t,s\right)-f_{X}\left(t\right)f_{Y}\left(s\right){\left|\right|}^{2}w\left(t,s\right)dtds}, \end{array}$$
where $f_{X}\left(\cdot \right),f_{Y}\left(\cdot \right),$ and $f_{X,Y}\left(\cdot \right)$ are the characteristic and joint characteristic functions of the random variables *X* (*p*-dimensional) and *Y* (*q*-dimensional). The weight function is given by $w\left(t,s\right)=(c_{p}c_{q}{\left||t|\right|}_{p+1}^{p}{\left||s|\right|}_{q+1}^{q}{)}^{-1}$, where $c_{d}=\pi^{(1+d)/2}/\Gamma \left(\left(1+d\right)/2\right)$. The calculation of dCov(X,Y) is more complex compared to the relatively simple calculations performed when computing the covariance for the CC. However, we are fortunate that the R package “energy”, authored by Rizzo, simplifies the calculation of the following definition. It is interesting to note that, according to [22], the population distance covariance coincides with the covariance with respect to Brownian motion, the random motion of particles suspended in a medium. In the same article, the distance correlation is described as the “natural extension” of the CC, and it is clear that the DC offers certain advantages over the CC.

In terms of advantages, DC surpasses CC in several ways. For example, while CC is restricted to two-dimensional variables, DC can handle variables in any dimension. Moreover, the range of R is between 0 and 1, which is inclusive. It is interesting to note that when CC =0, there is no linear correlation, but this does not indicate independence, whereas R(X,Y)=0 indicates independence between *X* and *Y*. Our aim is to utilize DC as a criterion for our filter method. However, having these advantages over CC does not necessarily mean that our filter method would perform better than the one presented in [20]. Nonetheless, there is a reason for optimism since [23] employs DC as a feature selection criterion in selecting features for energy polynomials. It is worth noting that they achieved a performance that matched that of the unfiltered models using two orders of magnitude fewer parameters.

### 2.1. Feature Selection Method in Random Forest

Our focus is on exploring how distance correlation can facilitate feature selection. To this end, we employ a feature selection algorithm to enhance our machine-learning models, particularly random forests (Algorithm 1). The goal of our feature selection algorithm is to reduce the feature space by considering the DC between each feature and the dependent variable, using a threshold value of R, denoted by $R^{*}$.

As outlined above, our approach involves creating a subset of this feature space using training data, which will then be employed to train a random forest model. To achieve this, we first specify a threshold value, denoted by $R^{*}$. We then compute $R(Y,X_{i})$ for i=1,…,p. Based on the resulting distance correlation values, we identify a subset of $X^{*}$, denoted by $X^{*}\subseteq X$, that includes any feature $X_{j}$ satisfying $R\left(Y,X_{j}\right)\geq R^{*}$. We subsequently employ $X^{*}$ to construct a random forest and compute the mean squared error (MSE) using test data.
**Algorithm 1** Proposed DC-based MethodGiven a training data set $D_{n}$ and the distance correlation set $\vec{R^{*}}$ of length *s*,Compute the distance correlation between *Y* and each feature $X_{j}$ and rank the features using the distance correlation.For each $R^{*}$,
(a)Eliminate the less correlated variables using the specified $R^{*}$ as a threshold.(b)Using the new training data with reduced feature space, construct a random forest using the Breiman RF algorithm.Given the *s* constructed random forests, select the model with the minimum prediction error based on the value of $R^{*}$.

### 2.2. Theoretical Results

In this section, we develop a large sample theory for the proposed DC-based feature selection method. We assume that our features are statistically independent and that only the relevant ones have a strong correlation with the response variable. Consider the model
$$\begin{array}{c} Y=f\left(X_{i}\right)+\epsilon_{i}. \end{array}$$

As in [19], we assume a moment condition on the random error terms $\epsilon_{i}$. Our goal is to ensure that our variable importance measure still converges and that it depends only on the filtered features. The *j*-th variable importance is calculated based on randomly permuting the values of $X_{j}$ in the out-of-bag sample, which is denoted by ${\tilde{X}}_{j}$. Given that we are using a regression tree and have chosen to minimize the sum of squared errors as our criterion, the resulting squared error after permutation can be calculated
$$\begin{array}{c} E_{{\tilde{X}}_{j}}{\left(Y-\hat{f}\left(X_{1},\ldots ,{\tilde{X}}_{j},\ldots ,X_{p^{*}}\right)\right)}^{2} \end{array}$$

We can express the variable importance for the *j*-th variable as follows.
$$\begin{array}{c} VI_{j}=\frac{E\left[{\left(f\left(X_{1},\ldots ,{\tilde{X}}_{j},\ldots ,X_{p}\right)-f\left(X_{1},\ldots ,X_{j},\ldots ,X_{p}\right)\right)}^{2}\right]}{E\left[{\left(Y-f\left(X_{1},\ldots ,{\tilde{X}}_{j},\ldots ,X_{p}\right)\right)}^{2}\right]}. \end{array}$$

**Theorem 1.** 
*Under assumptions 3.1, 3.2, 3.3, and 3.4 of [19], and there exists a fixed constant 1<B<∞, for any κ>0, the estimated variable importance converges to the true variable importance at an exponential rate. That is*

$$\begin{array}{c} P\left(|{\hat{VI}}_{j}-VI_{j}|>\kappa \right)\leq e^{-\kappa \cdot n^{\nu (p^{*})}/B}, \end{array}$$

*where $0<v(p^{*})\leq 1$ is a function of the dimension $p^{*}$, which represents the reduced number of features obtained using the DC-based filter method. $VI_{j}$ is a measure of variable importance for each variable j∈P, as defined in (Section 2.2), along with its estimate ${\hat{VI}}_{j}$.*


**Proof.** Employing analogous reasoning as presented in [20], we can establish the validity of Theorem 1. Consequently, the detailed proof is omitted here. □

## 3. Simulation Study

In this section, we perform a simulation study to assess the efficacy of our proposed method. In addition to the simulation setup used in [20], we examine two additional settings. For each setting, we generate 200 training samples and 1000 test samples. We evaluate the performance of our approach for various numbers of features, namely p=80,100,300,500.

Under settings 1 & 2, we consider the following model
$$\begin{array}{c} \mathrm{Model}\;1:\;\;Y_{i}=5\left(X_{i,1}+X_{i,2}+X_{i,3}+X_{i,4}\right)+\epsilon_{i} \end{array}$$
where $\epsilon_{i}$’s are the random errors that are normally distributed with a mean of 0 and variance of 1.
–**Setting 1:** Generate $X_{i}$ from a normal distribution: $N\left(0_{p\times 1},\Sigma_{p\times p}\right)$, where $\Sigma_{i,j}=\rho^{\left|i-j\right|}$, with ρ=0.5 and 0.8.–**Setting 2:** Generate $X_{i}$ from a normal distribution: $N\left(0_{p\times 1},\Sigma_{p\times p}\right)$, where $\Sigma_{i,j}=\rho^{\left|i-j\right|}+0.2I_{\left(i\neq j\right)}$, with ρ=0.5Under setting 3, we consider the following model
$$\begin{array}{c} \mathrm{Model}\;2:\;\;Y_{i}=X_{i,1}^{2}+X_{i,20}+X_{i,33}^{3}+X_{i,55}^{2}+\epsilon_{i} \end{array}$$
where $\epsilon_{i}$’s are the random errors that are normally distributed with a mean of 0 and variance of 1.
–**Setting 3:** Generate $X_{i}$ from a normal distribution: $N\left(0_{p\times 1},\Sigma_{p\times p}\right)$, where $\Sigma_{i,j}=\rho^{\left|i-j\right|}$, with ρ=0.8.Under setting 4, we consider the following model
$$\begin{array}{c} \mathrm{Model}\;3:\;\;Y_{i}=100\times {\left(X_{i,1}-0.5\right)}^{2}\times {\left(X_{i,2}-0.25\right)}^{+}+\epsilon_{i} \end{array}$$
where ${\left(\cdot \right)}^{+}$ represents the positive part and $\epsilon_{i}$’s are the random errors that are normally distributed with a mean of 0 and variance of 1.
–**Setting 4:** Generate $X_{i}$ from $Unif{\left[0,1\right]}^{p}$.

The first step of our method involves calculating the distance correlation between the response variable *Y* and each feature variable $X_{j}$ for all j=1,…,p. Next, we use pre-defined thresholds to select significant features. These thresholds are determined based on minimum distance correlation levels between *Y* and $X_{j}$, which include ${\vec{R}}^{*}=\left\{0.00,0.10,0.20,0.30,0.40,0.50,0.60\right\}$. If $R^{*}=0$, then all features are selected and included in the random forest regression. Conversely, if $R^{*}=0.5$, then only features with a distance correlation of at least 0.5 with the response variable are selected and added to the RF at each stage. We repeated the procedure 200 times to obtain reliable results.

### 3.1. Analysis of the Linear Models

Table 1 presents the results for all methods for Model 1 and setting 1 with ρ=0.5.

One trend that is evident is that the increase in the number of parameters (*p*) leads to an increase in the MSE. This implies that the model’s accuracy decreases as the number of parameters increases, which is expected. The RLTNo5 model, which is RLT without muting where five features are utilized in the linear combination to create a split candidate, performed significantly better than other models. On the other hand, the traditional RF had the worst performance, which is desirable since our aim is to enhance the traditional RF with our methods. The optimal $r^{*}$ threshold is likely between 0.4 and 0.6, although the optimal $R^{*}$ threshold value is inconclusive. Nonetheless, the general trend indicates that as $R^{*}$ increases, MSE decreases. It appears that the best model has an $R^{*}>0.6$, but we found that this was not the case. For $R^{*}>0.6$, the model’s accuracy decreased, and we even encountered errors for $R^{*}$ values that were excessively high since this meant that the model was discarding all parameters, and as a result, no random forest could be generated. It is probable that for these settings, the optimal $R^{*}$ threshold is between 0.5 and 0.7.

We observed a significant improvement in the CC method’s performance in the RF model when $r^{*}$ increased from 0.1 to 0.2 in the p=500 column. This resulted in a 44.7% decrease in MSE. Similarly, there was a 45.4% reduction in MSE when our method’s threshold $R^{*}$ increased from 0.4 to 0.5. It is possible that the similarity in the magnitude of these MSE drops is coincidental. However, we observed a similar pattern for p=80,100, and 300. To clarify, let ${\mathrm{MSE}}_{{\mathrm{DC}}_{R^{*},p}}$ represent the DC MSE at $R^{*}$ and *p*. Similarly, let ${\mathrm{MSE}}_{{\mathrm{CC}}_{r^{*},p}}$ be the CC MSE at $r^{*}$ and *p*. We noticed the following trend:$$\begin{array}{cc} \left|\frac{{\mathrm{MSE}}_{{\mathrm{DC}}_{0.5,80}}}{{\mathrm{MSE}}_{{\mathrm{DC}}_{0.4,80}}}-\frac{{\mathrm{MSE}}_{{\mathrm{CC}}_{0.2,80}}}{{\mathrm{MSE}}_{{\mathrm{CC}}_{0.1,80}}}\right| & =0.0774\\ \left|\frac{{\mathrm{MSE}}_{{\mathrm{DC}}_{0.5,100}}}{{\mathrm{MSE}}_{{\mathrm{DC}}_{0.4,100}}}-\frac{{\mathrm{MSE}}_{{\mathrm{CC}}_{0.2,100}}}{{\mathrm{MSE}}_{{\mathrm{CC}}_{0.1,100}}}\right| & =0.0656\\ \left|\frac{{\mathrm{MSE}}_{{\mathrm{DC}}_{0.5,300}}}{{\mathrm{MSE}}_{{\mathrm{DC}}_{0.4,300}}}-\frac{{\mathrm{MSE}}_{{\mathrm{CC}}_{0.2,300}}}{{\mathrm{MSE}}_{{\mathrm{CC}}_{0.1,300}}}\right| & =0.0327\\ \left|\frac{{\mathrm{MSE}}_{{\mathrm{DC}}_{0.5,500}}}{{\mathrm{MSE}}_{{\mathrm{DC}}_{0.4,500}}}-\frac{{\mathrm{MSE}}_{{\mathrm{CC}}_{0.2,500}}}{{\mathrm{MSE}}_{{\mathrm{CC}}_{0.1,500}}}\right| & =0.0065 \end{array}$$

The DC-based model accuracy eventually improves to a comparable level with the CC-based model when $R^{*}$ reaches approximately 0.5. However, this is not the optimal $R^{*}$ value, just as $r^{*}=0.2$ is not the optimal threshold. In this case, the CC method easily identifies the more important parameters, while the DC method is more cautious and does not filter out parameters with weak linear correlations. The best prediction MSEs are achieved at $r^{*}=0.5$ for the CC method and $R^{*}=0.6$ for the DC method. Although a higher $R^{*}$ threshold is required for the DC method to optimize, the prediction MSE results are comparable to those of the CC method.

According to Table 2, we see the same optimal threshold values of $r^{*}$ and $R^{*}$. The optimal MSEs for the DC and CC methods are even closer, but the CC method still has a slight edge. The race for the best MSE is now closer with RLT, but RLTNo5 remains the best model, while the traditional RF remains the least accurate. As the correlation between parameters and the response variable increases, the MSE generally decreases compared to Table 1.

In Table 3, we observe that the CC method outperforms our method and marks the first instance where a better model than RLTNo5 is identified. It is possible that the DC method could achieve comparable results at a higher threshold, but we did not have the opportunity to optimize this threshold for the DC method.

### 3.2. Analysis of the Nonlinear Model

In this section, we examine a nonlinear model as outlined in setting 3. The results are presented in Table 4.

These results are particularly exciting as they reveal the advantages of using DC as a feature selection criterion. It is worth noting that the CC method threshold stops at 0.3 because, as the data are not constructed under a linear model, setting a CC threshold higher than 0.3 will filter out all the parameters of the model, making it impossible to construct an RF. This is not the case with the DC method, as it is capable of detecting nonlinear correlations and allowing more parameters to survive the filter method. Although the CC method does not perform well in this case, we can see that RLT remains the best method for p=80,100, and 300. However, for the high-dimensional case, our proposed method performs best, indicating that it could be an improvement over RF in high-dimensional scenarios. In the future, it would be interesting to compare the proposed method with other machine learning techniques in high-dimensional datasets that exhibit nonlinear correlations. Additionally, we assess the benefits of the proposed method using the simulation setting employed in a previous study [19]. The outcomes of this analysis are presented in Table 5.

The performance of our DC method is outstanding compared to both traditional RF and the CC method in this nonlinear simulated dataset, similar to our other nonlinear simulated dataset. It is worth noting that the CC method has a lower threshold, as a threshold higher than 0.3 eliminates all parameters from the RF model. As we have previously observed, RLT performs exceptionally well here. However, as seen in setting 3, as the number of parameters increases, our proposed method appears to gain an advantage over RLT. Specifically, the DC-based feature selection method outperforms RLT for p=300 and p=500. This once again supports the notion that the DC-based method may be an excellent candidate for high-dimensional data analysis.

According to Figure 1, it is evident that the DC-based method outperforms CC significantly. In setting 4, we observe that our optimal MSE is often less than half of the CC method’s MSE.

## 4. Applications

To illustrate the practical usage, we apply our proposed methods to two real datasets, which are provided below.


**Riboflavin Data:**
This dataset contains riboflavin production by Bacillus subtilis. There are n=71 observations of p=4088 predictors (gene expressions) and a one-dimensional response variable.
**Boston Housing Data:**
This dataset contains housing data for 506 census tracts of Boston from the 1970 census. There are n=506 observations of p=14 predictors.

### 4.1. Riboflavin Data

The Riboflavin dataset is a widely used dataset found in the ‘hdi’ R package, provided by [24]. It consists of 71 observations of 4088 predictors, representing the expression levels of 4088 genes, and a single response variable, which is the riboflavin production of Bacillus Subtilis. The objective of our study is to predict the log-transformed riboflavin production rate using gene expressions as predictors. This dataset is an example of a high-dimensional dataset, as the number of features is much larger than the number of observations, i.e., p>n. The results of our analysis are presented in Table 6.

To ensure stable results, we conduct 200 repetitions and calculate the average prediction mean squared error. The findings indicate that the CC-based feature selection method is much more precise than the RLT methods and significantly better than the traditional RF. Our proposed method comes in second place with an optimal threshold of 0.7. It is worth noting that a better $R^{*}$ threshold may exist in the range of (6.5,7.5).

The results indicate that the methods have similar accuracy, but the CC method performs better. In support of this, Figure 2 shows a continued decrease in MSE as the $R^{*}$ threshold increases, suggesting that an optimal threshold may exist beyond 0.7. However, even with this potential for improvement, the results obtained with our proposed method are comparable at best to those of the CC method.

Figure 3 illustrates the diminishing returns of increasing the CC threshold and highlights the potential for a better prediction of mean squared error (MSE) by increasing the DC threshold.

### 4.2. Boston Housing Data

The Boston housing data set is provided by [25] and is a built-in data set in R. Unlike the riboflavin data set, it has a lower dimensionality with only 13 predictors and a one-dimensional response variable. The data set contains 506 observations and provides information gathered from the 1970s census. The predictors include the per capita crime rate by town, the average number of rooms per dwelling, the pupil-teacher ratio by town, and other factors. The response variable is the median value of owner-occupied homes in $1000. The objective is to use the available information, such as the per capita crime rate by town (CRIM), nitric oxides concentration (NOX), proportion of non-retail business acres per town (INDUS), and full-value property-tax rate per $10,000 (TAX), among others, to predict the median value of owner-occupied homes.

We applied the same methodology to analyze the Boston housing dataset, and the prediction MSE results are presented in Table 7. Similar to the Riboflavin dataset, we do not observe any improvement in the model by using the RLT method. However, we see slight improvements from the two filter methods compared to traditional RF. Furthermore, we notice that our proposed method slightly outperforms the CC method. Moreover, we observe that our proposed DC-based method has relatively stable results irrespective of the $R^{*}$, whereas the CC method shows an increasing trend in prediction MSE and results in almost three times the MSE of the traditional RF as $r^{*}$ varies from 0.1 to 0.7.

Once again, the results obtained support the notion that our DC-based feature selection method is more conservative in eliminating predictors that are relevant to the RF model compared to other methods. It is interesting to note that a similar trend as seen in Figure 4 can also be observed in Figure 5, where the change in MSE of the CC method from $r^{*}=0.2$ to 0.4 is similar to that of the DC method from $R^{*}=0.5$ to 0.7. This change in MSE is approximately 12% for both methods, as $r^{*}$ and $R^{*}$ vary within those ranges.

## 5. Conclusions

In this paper, we proposed a novel variable selection procedure for RF using distance correlation. We observed that the proposed DC-based method performed very well in most cases, especially in nonlinear models. Although we anticipated that our approach would perform similarly or better than the CC-based filter method, we were pleasantly surprised to find that it outperformed RLT methods under high-dimensional settings. Our approach consistently outperformed the traditional RF method, and in the case of the nonlinear models, it even outperformed the CC method. In the linearly simulated data, we observed that the DC method performed similarly to the CC method in most cases. However, we noticed that optimizing the DC prediction MSE required a higher threshold, which is not surprising given that our method is more conservative in feature filtering. This is not a significant disadvantage, except perhaps for computational cost, as more features are retained in the RF model construction. To address this, we can adjust the threshold to a higher value. We observed only one case where DC significantly underperformed the CC method, which was in setting 2. In this case, a strong linear correlation was simulated, and thus, the CC method was expected to perform well, which was indeed the case. However, in situations such as this, we can consider increasing the DC threshold to 0.7 or 0.8 and see if the prediction MSE improves and becomes comparable to that of the CC method, as we observed previously. Our method demonstrated its superior performance in nonlinear models, particularly in high-dimensional cases. This piqued our interest in exploring high-dimensional datasets. Finally, two real data applications are provided to illustrate the advantage of the proposed methods.

## Figures and Tables

**Figure 1 entropy-25-01250-f001:**
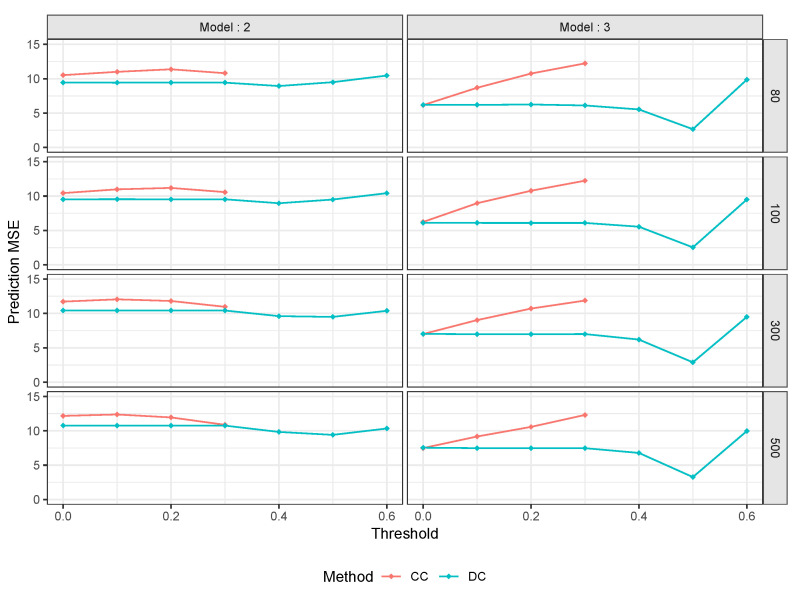
Prediction MSE Comparison for Model 2 & 3.

**Figure 2 entropy-25-01250-f002:**
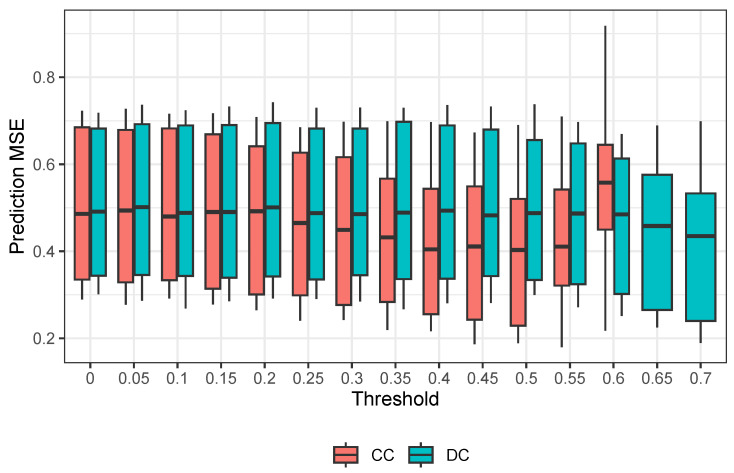
Boxplot for Prediction MSE Comparison for Riboflavin Data.

**Figure 3 entropy-25-01250-f003:**
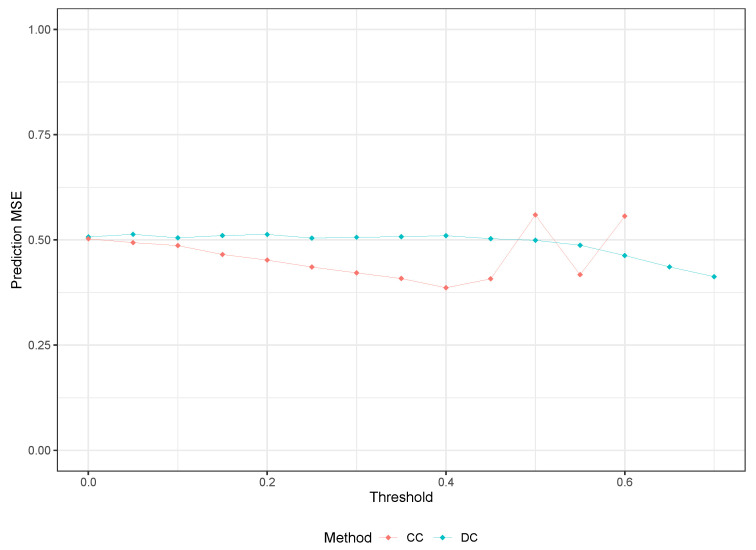
Prediction MSE Comparison for Riboflavin Data for CC and DC-based Methods.

**Figure 4 entropy-25-01250-f004:**
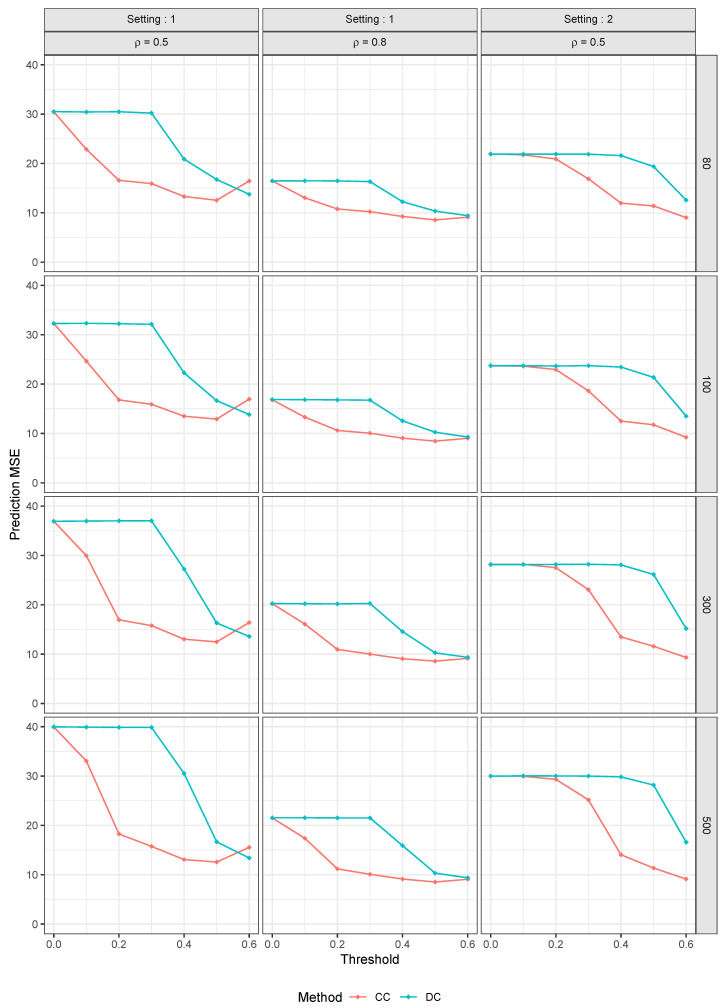
Prediction MSE Comparison for Setting 1 (ρ = 0.5, 0.8) and Setting 2 with ρ=0.5.

**Figure 5 entropy-25-01250-f005:**
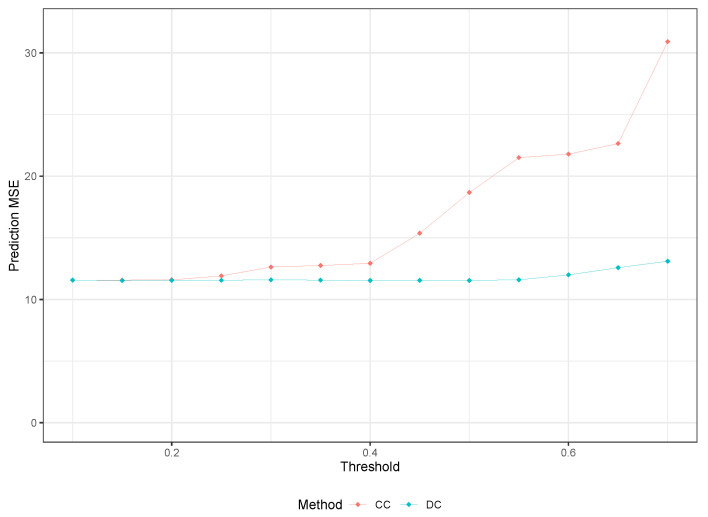
Prediction MSE Comparison for Boston Housing Data for CC and DC-based Methods.

**Table 1 entropy-25-01250-t001:** Prediction Mean Squared Error for Model 1 and Setting 1 with ρ=0.5.

Method		p=80	p=100	p=300	p=500
**Traditional RF**		**30.4468**	**32.3146**	**37.0157**	**39.9092**
No	RLTNo1	17.1149	18.2449	20.6827	22.2395
	RLTNo2	8.3586	9.2965	10.8636	12.1497
	RLTNo5	5.9539	6.8420	8.4067	9.5437
Moderate	RLTMod1	23.5688	24.9247	29.2962	31.4494
	RLTMod2	12.7399	13.8862	16.9476	19.1914
	RLTMod5	9.7806	10.9047	13.5140	15.6142
CC $\left(r^{*}\right)$	0	30.4568	32.3099	36.9560	39.9454
	0.1	22.8696	24.6372	29.9454	33.0442
	0.2	16.5787	16.7887	16.9566	18.2652
	0.3	15.9218	15.8904	15.7830	15.7455
	0.4	13.3106	13.4890	13.0326	13.0766
	0.5	12.5500	12.8932	12.4917	12.5678
	0.6	16.4444	16.9558	16.4051	15.5541
DC $\left(R^{*}\right)$	0	30.5103	32.2662	36.9264	39.9739
	0.1	30.4394	32.3129	36.9792	39.9157
	0.2	30.4860	32.2304	37.0245	39.8639
	0.3	30.2126	32.1138	37.0334	39.8655
	0.4	20.8794	22.2660	27.2499	30.5149
	0.5	16.7517	16.6341	16.3208	16.6678
	0.6	13.7511	13.8123	13.5889	13.3938

**Table 2 entropy-25-01250-t002:** Prediction Mean Squared Error for Model 1 and Setting 1 with ρ=0.8.

Method		p=80	p=100	p=300	p=500
**Traditional RF**		**16.4542**	**16.8286**	**20.2293**	**21.4920**
No	RLTNo1	11.1426	11.6650	13.5729	14.1749
	RLTNo2	6.8722	7.3101	8.8551	9.6527
	RLTNo5	5.4821	5.8649	7.3025	8.0649
Moderate	RLTMod1	14.9992	15.5370	18.7807	19.8693
	RLTMod2	10.3251	10.8486	13.8485	15.1718
	RLTMod5	8.4156	8.9015	11.3316	12.5533
CC $\left(r^{*}\right)$	0	16.4618	16.8028	20.2333	21.5206
	0.1	13.0510	13.2847	16.1036	17.3913
	0.2	10.7760	10.5976	10.9608	11.1928
	0.3	10.2295	10.0385	10.0109	10.0872
	0.4	9.2580	9.0398	9.0732	9.1315
	0.5	8.5590	8.4243	8.5828	8.5259
	0.6	9.1113	9.0128	9.1327	9.0838
DC $\left(R^{*}\right)$	0	16.4589	16.8685	20.2596	21.5370
	0.1	16.4747	16.8312	20.2180	21.5444
	0.2	16.4707	16.7899	20.1973	21.5172
	0.3	16.3218	16.7368	20.2653	21.5056
	0.4	12.2518	12.5301	14.5710	15.9063
	0.5	10.3558	10.2450	10.2731	10.3228
	0.6	9.4236	9.2640	9.3533	9.3839

**Table 3 entropy-25-01250-t003:** Prediction Mean Squared Error for Model 1 and Setting 2 with ρ=0.5.

Method		p=80	p=100	p=300	p=500
**Traditional RF**		**21.9640**	**23.6652**	**28.2053**	**30.0032**
No	RLTNo1	13.0988	14.2620	16.5793	17.3747
	RLTNo2	7.3378	8.2417	10.2177	11.1712
	RLTNo5	5.5720	6.3689	8.2305	9.2038
Moderate	RLTMod1	17.9596	19.3122	23.0986	24.3520
	RLTMod2	11.4233	12.5715	16.2147	17.8654
	RLTMod5	9.1465	10.2833	13.5496	15.2372
CC $\left(r^{*}\right)$	0	21.9342	23.6987	28.1940	29.9885
	0.1	21.7451	23.6321	28.2193	29.9617
	0.2	20.9032	22.9340	27.5293	29.3341
	0.3	16.8882	18.6162	23.0721	25.1728
	0.4	11.9670	12.4959	13.4938	14.0448
	0.5	11.3873	11.7433	11.6022	11.3566
	0.6	9.0305	9.2198	9.3215	9.1254
DC $\left(R^{*}\right)$	0	21.9021	23.7338	28.1547	29.9792
	0.1	21.8892	23.7192	28.1623	30.0492
	0.2	21.8888	23.6486	28.1887	30.0208
	0.3	21.8853	23.7239	28.2238	29.9949
	0.4	21.6011	23.4470	28.0920	29.8334
	0.5	19.3799	21.3558	26.1481	28.1744
	0.6	12.5753	13.4929	15.1863	16.6041

**Table 4 entropy-25-01250-t004:** Prediction Mean Squared Error for Model 2 and Setting 3 with ρ=0.8.

Method		p=80	p=100	p=300	p=500
**Traditional RF**		**9.4389**	**9.5245**	**10.4246**	**10.7869**
No	RLTNo1	8.6755	8.7385	9.4071	9.7955
	RLTNo2	8.5479	8.6631	9.4587	9.9032
	RLTNo5	8.6720	8.7762	9.5994	10.0118
Moderate	RLTMod1	9.6584	9.7615	10.7009	11.2133
	RLTMod2	9.7378	9.8579	10.9569	11.4871
	RLTMod5	9.8222	9.9758	11.0402	11.6132
CC $\left(r^{*}\right)$	0	10.5241	10.4354	11.7246	12.1731
	0.1	11.0046	10.9849	12.0554	12.3790
	0.2	11.3745	11.1895	11.8162	11.9509
	0.3	10.8041	10.5800	10.9673	10.8763
DC $\left(R^{*}\right)$	0	9.4371	9.5271	10.4387	10.7732
	0.1	9.4270	9.5461	10.4322	10.7692
	0.2	9.4465	9.5276	10.4433	10.7636
	0.3	9.4336	9.5344	10.4295	10.7577
	0.4	8.9385	8.9611	9.6091	9.8364
	0.5	9.4990	9.4992	9.5010	9.4111
	0.6	10.4607	10.4244	10.3874	10.3362

**Table 5 entropy-25-01250-t005:** Prediction Mean Squared Error for Model 3 and Setting 4.

Method		p=80	p=100	p=300	p=500
**Traditional RF**		**6.1719**	**6.3132**	**7.0491**	**7.4381**
	RLTNo1	2.4868	2.4958	2.9554	3.3648
	RLTNo2	2.5882	2.6486	3.3094	3.8033
	RLTNo5	2.8512	2.8675	3.5907	4.3271
	RLTMod1	3.1720	3.1258	3.8918	4.5346
	RLTMod2	3.6176	3.5186	4.5701	5.1221
	RLTMod5	3.7851	3.7519	4.8743	5.7918
CC $\left(r^{*}\right)$	0	6.1638	6.2397	7.0040	7.4891
	0.1	8.6832	8.9644	9.0353	9.1730
	0.2	10.7540	10.7789	10.7112	10.5731
	0.3	12.2340	12.2444	11.8764	12.3109
DC $\left(R^{*}\right)$	0	6.1879	6.1030	7.0218	7.5451
	0.1	6.1925	6.0984	6.9839	7.4811
	0.2	6.2513	6.0910	6.9863	7.4811
	0.3	6.1112	6.0962	7.0018	7.4744
	0.4	5.5324	5.5445	6.2003	6.7826
	0.5	2.6557	2.5385	2.8895	3.2704
	0.6	9.8633	9.5040	9.4988	9.9643

**Table 6 entropy-25-01250-t006:** Prediction Mean Squared Error for Riboflavin Data.

Traditional RF		0.5029
No	RLTNo1	0.5521
	RLTNo2	0.5459
	RLTNo5	0.5436
Moderate	RLTMod1	0.5555
	RLTMod2	0.5216
	RLTMod5	0.5623
Threshold	CC $\left(r^{*}\right)$	DC $\left(R^{*}\right)$
0.00	0.5026	0.5071
0.05	0.4936	0.5133
0.10	0.4866	0.5049
0.15	0.4654	0.5104
0.20	0.4521	0.5130
0.25	0.4356	0.5043
0.30	0.4217	0.5063
0.35	0.4083	0.5076
0.40	0.3864	0.5100
0.45	0.4076	0.5029
0.50	0.5594	0.4990
0.55	0.4175	0.4873
0.60	0.5565	0.4628
0.65	NA	0.4358
0.70	NA	0.4126

**Table 7 entropy-25-01250-t007:** Prediction Mean Squared Error for Boston Housing Data.

Traditional RF		11.6123
No	RLTNo1	16.5492
	RLTNo2	16.7430
	RLTNo5	16.0898
Moderate	RLTMod1	16.0028
	RLTMod2	15.6108
	RLTMod5	15.6015
Threshold	CC $\left(r^{*}\right)$	DC $\left(R^{*}\right)$
0.1	11.5548	11.5702
0.15	11.5674	11.5258
0.2	11.5926	11.5477
0.25	11.9115	11.5586
0.3	12.6297	11.5891
0.35	12.7505	11.5651
0.4	12.9315	11.5344
0.45	15.3672	11.5441
0.5	18.6801	11.5417
0.55	21.5029	11.5951
0.6	21.7865	11.9905
0.65	22.6410	12.5806
0.7	30.9052	13.0999

## Data Availability

Data available in a publicly accessible repository.

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
