# Peer review of "Distance Correlation-Based Feature Selection in Random Forest"

_entropy, 2023, doi:10.3390/e25091250_

Round 1
Reviewer 1 Report
The authors propose a new approach for building predictive models by combining distance correlation screening/thresholding and random forest. Some theoretical justification is provided. Numerical studies demonstrate the superior performance of the proposed method over traditional methods, such as no screening and screening via Pearson correlation. In my opinion, the proposed methodology is relatively straightforward, which reduces the novelty of the paper. Below are my detailed comments.
In practice, it is common to see analysts performing a variable screening before building machine learning models. According to the simulation, CC performs better under the linear case while DC under the nonlinear case. In practice, it is hard to know for sure whether the data exhibits linearity or nonlinearity. Therefore, why not do screening using both Pearson and distance correlations? Overall, I find the results in the numerical studies not entirely convincing.
How do analysts select the threshold parameter R*? Apparently, the performance of DC highly depends on the choice of R*. The authors should consider providing some guidelines for selecting such a tuning parameter.
Reviewer 2 Report
This paper aims to address the challenges posed by high-dimensional data and feature selection in traditional Random Forest (RF) algorithms. The authors propose a method incorporating distance correlation (dCor) into Random Forest regression for feature selection. dCor, a metric able to capture all types of dependencies between random vectors of arbitrary dimensions, is utilized as a preprocessing step in the RF algorithm. This assists in selecting features significantly correlated to the response variable. Through simulations and real-world applications, the authors demonstrate that their approach is competitive and often superior in handling high-dimensional nonlinear datasets.
- The proposition of applying distance correlation prior to implementing random forest is not entirely novel. Other studies (e.g., https://arxiv.org/pdf/2006.12919.pdf and https://egrove.olemiss.edu/etd/1800/?utm_source=egrove.olemiss.edu%2Fetd%2F1800&utm_medium=PDF&utm_campaign=PDFCoverPages) have explored this approach. Therefore, the authors should clearly articulate what differentiates their proposal from the existing literature.
- The threshold R* selection is a sensitive aspect of the model. A poor selection may negatively impact the performance of the random forest. It would be beneficial if the authors could discuss whether this crucial value can be determined independently of the out-of-bag error.
- Notably, in most experiments presented in this paper, the RF models appear to underperform compared to the RLT models. This suggests that exploring more complex models could better demonstrate the necessity and advantages of using random forests. The authors might wish to delve into this aspect further.
Round 2
Reviewer 2 Report
My concerns have been addressed.